# Structural basis for the SUMO protease activity of the atypical ubiquitin-specific protease USPL1

Ying Li[1], Nathalia Varejão [1] & David Reverter [1]✉

Post-translational protein modifications by ubiquitin and ubiquitin-like modifiers regulate many major pathways in the cell. These modifications can be reversed by de-ubiquitinating enzymes such as ubiquitin-specific proteases (USPs). Proteolytic activity towards ubiquitin-modified substrates is common to all USP family members except for USPL1, which shows a unique preference for the ubiquitin-like modifier SUMO. Here, we present the crystal structure of USPL1 bound to SUMO2, defining the key structural elements for the unusual deSUMOylase activity of USPL1. We identify specific contacts between SUMO2 and the USPL1 subdomains, including a unique hydrogen bond network of the SUMO2 C-terminal tail. In addition, we find that USPL1 lacks major structural elements present in all canonical USPs members such as the so-called blocking loops, which facilitates SUMO binding. Our data give insight into how a structural protein scaffold designed to bind ubiquitin has evolved to bind SUMO, providing an example of divergent evolution in the USP family.

[1] Institut de Biotecnologia i de Biomedicina (IBB) and Dept. de Bioquímica i Biologia Molecular, Universitat Autònoma de Barcelona, 08193 Bellaterra, Spain.
✉email: david.reverter@uab.cat

Small ubiquitin-like modifiers (SUMO) belong to the family of ubiquitin-like proteins (UbLs), which are post-translational modifiers that regulate a wide plethora of protein functions in cells[1,2]. Among them, the most preeminent function of ubiquitin is the degradation of intracellular proteins by the ubiquitin-proteasome system (UPS)[3]. SUMOylation contributes to many cellular pathways and is involved in many functions, such as DNA replication, nuclear transport, or DNA damage control[4–6]. SUMO, ubiquitin, and all other UbL modifiers are covalently attached to target proteins by the formation of an isopeptide bond with an internal lysine residue, but before that, UbLs need to be activated by a dedicated conjugation pathway through an enzymatic cascade formed by E1, E2, and E3 enzymes[7,8]. The covalent attachment of ubiquitin and SUMO to protein targets is a reversible process through the action of de-ubiquitinating (DUBs) or de-SUMOylating proteases[9,10].

DUBs can be divided into seven families, namely the Ubiquitin C-terminal hydrolases (UCHs), Ubiquitin-specific proteases (USPs), Machado–Joseph Disease protease family (MJDs), Ovarian tumor proteases (OTUs), MINDY protease family, JAMM family, and ZUFSP/Mug105 family, each of which presents a unique structural fold[11–14]. Most DUB families are cysteine proteases, except the JAMM family, which are metalloproteases. USPs constitute the largest DUB family constituted by 56 members in humans. The catalytic domain of USPs is composed of three subdomains, named thumb, palm, and fingers, and the overall structure resembles the shape of a human right hand[15,16]. The active site of USPs is formed by a catalytic triad (Cys-His-Asp/Asn) located between the palm and thumb subdomains, which together with the fingers subdomain grasps ubiquitin for proteolytic cleavage[15]. In addition, USPs are multidomain enzymes in which adjacent domains aid to achieve multiple functions, such as zinc finger domain (ZnF), ubiquitin-like domain (UbL), ubiquitin-related domain (UBA) or ubiquitin interaction motif (UIM)[11,16]. USPs are active towards ubiquitin, but there are two exceptions: USP18, which is active towards the double-headed ISG15; and USPL1, a distant member of the family specific for SUMO[17,18]. Whilst ISG15 and ubiquitin share a 30% sequence identity and the structural analysis showed a quite similar binding interface with USP18, SUMO, and ubiquitin display little homology (16% sequence identity), foreseeing the presence of a divergent binding mechanism between USPL1 and SUMO.

Until recently, SUMO proteases were only constituted by the six members of the SENP protease family in humans, which are similar to ULP1, the first discovered SUMO protease in *S. cerevisiae*[19] and all belong to the C48 cysteine protease structural class (SENP1-SENP3 and SENP5-SENP7)[10]. However, two novel types of SUMO proteases have been described in recent years without sequence/structural homology to the SENP/ULP family: the deSUMOylating peptidase 1 and 2 (DESI1 and DESI2)[20]; and the deubiquitinating enzyme USPL1, which is a member of the ubiquitin USP family but is specific for SUMO, instead of ubiquitin[18]. An analogous finding was observed in the characterization of SENP8/DEN1, which is another member of the SUMO protease family (SENP/ULP family), but specific for the Nedd8, another type of UbL modifier[21–24].

Full-length of USPL1 is composed of 1.092 residues containing a USP-like catalytic domain in a middle region and long, disordered protein extensions without the presence of evident globular domains. So far, the only function described for USPL1 takes place in the nucleus, where it is a component of the Cajal Bodies (CBs), co-localizing with coilin[18]. USPL1 seems to have an impact on the formation and dynamics of CBs and in cell proliferation, as observed in the USPL1-depleted cells[18]. Whilst USPL1 deletion does not affect the overall SUMOylation in the cell, it causes significant frizzled protein mislocalization and damage in cell proliferation, which interestingly does not depend on its deSUMOylase catalytic activity[25]. The CBs are membrane-less compartments involved in the biogenesis of snRNP and snoRNP, maintenance of telomeres, and processing of histone mRNA[26]. USPL1 is a low-abundant component of the CBs that plays a role in the RNAPII transcription of snRNA that is essential for cell growth[25]. Moreover, SUMO has been involved in tumorigenesis, genetic variants of USPL1 are closely related to grade-3 breast cancer[27], as well as USPL1 may also be involved in the signaling pathway of multiple myeloma[28].

USPL1 can interact with both SUMO1 and SUMO2/3, but it shows a much higher activity for the SUMO2/3 isoform[18]. The sequence of the catalytic domain of USPL1 is unequivocally related to USP members with deubiquitinase activity, probably forming the canonical right hand-like structure with palm, thumb, and fingers subdomains. The existing understanding of USPL1 is not abundant, and the determinants of the particular specificity for SUMO have not yet been determined. In order to get insights into the interaction of USPL1 with SUMO, here we show the crystal structure of the complex between human USPL1 and the human precursor of SUMO2 at 1.8 Å resolution. Structural and biochemical analysis of the complex interface of USPL1-SUMO2 by mutagenesis analysis sheds light on key structural determinants of this unusual USP family member that has evolved to be specific for SUMO.

## Results

**Covalent crosslinked complex between USPL1 and SUMO2 precursor.** The catalytic domain for recombinant expression of human USPL1 is based on the sequence alignment with members of the USP ubiquitin-specific protease family and comprises residues Met212 through Leu514 (Supplementary Fig. 1)[18]. The sequence identity of the catalytic domain among members of the USP family is typically around 15–50% and most are specific for cleaving off ubiquitin from protein targets or ubiquitin chains (or ISG15 in the case of USP18)[16]. However, USPL1 sequence identity is one of the lowest in the USP family[14], with identities ranging from 16% for USP7 to 14% for USP28, as examples of two USP members. The homology is basically observed in the secondary structure elements around the catalytic triad of the active site (Cys236-His456-Asp472) (Supplementary Fig. 1). This sequence divergence with all members of the USP family reflects the existing relevant substitutions in USPL1, which evolved to cleave off SUMO rather than ubiquitin from protein targets.

The complex between USPL1 and SUMO2 was prepared by taking advantage of the formation of a covalent bond between the SUMO2 C-terminal glycine (Gly93) and the active site cysteine of USPL1 (Cys236), which stabilizes the complex, increases the binding affinity and thus the chances of crystallization. To prepare this covalent bond, we used the dehydroalanine (DHA) strategy[29], which creates an electrophilic center highly reactive with nucleophiles such as internal cysteines in proteins (Fig. 1a). First, in the human SUMO2 precursor, Gly93 was substituted for cysteine, and Cys48 was substituted for serine, leaving only one internal cysteine at the position of the reactive Gly93. Next, the SUMO2-C48S/G93C precursor was incubated with 2,5-dibromo-hexanediamide, which can desulfurize the Cys93 of the SUMO precursor into DHA at mild conditions. Finally, the SUMO2 precursor bearing DHA at position of Gly93 was incubated with USPL1 to create a covalent bond between active site nucleophile cysteine of USPL1 and the DHA electrophilic trap on the SUMO2 precursor (Fig. 1a).

A major concern of the DHA strategy in SUMO proteases is the possible steric hindrances in the C-terminal di-glycine motif

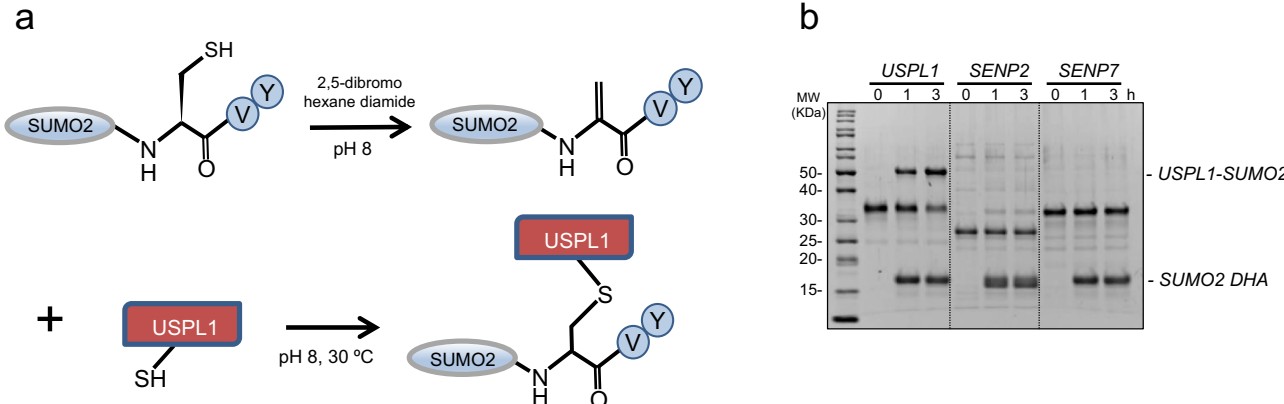

**Fig. 1 Covalent thioether bond formation of SUMO2-DHA with USPL1 catalytic domain. a** Schematic representation of the reaction to form the covalent thioether between the SUMO2-C48S/G93C precursor and USPL1 catalytic domain. **b** Active site probe assay for USPL1, SENP2, and SENP7 with SUMO2-DHA. Time course assay of USPL1, SENP2, and SENP7 at 2 μM using the SUMO2-DHA substrate at 6 μM at 30 °C for 3 h. n = 3 technical replicates. Source data are provided as a Source Data file.

after the substitution of Gly93 for DHA, which might clash with narrow protease binding pockets. Therefore, we initially checked the formation of the covalent crosslink after incubation of SUMO2 DHA precursor with USPL1, and two other well-characterized SUMO proteases, SENP2 and SENP7 (Fig. 1b). Interestingly, only USPL1 is able to form the covalent crosslink with the SUMO2 DHA precursor, indicating a correct and specific binding between SUMO2 and USPL1. However, the two other well-characterized SUMO proteases, SENP2 and SENP7, are unable to form a covalent bond between the active site cysteine and DHA (Fig. 1b). As known in the SENP/ULP protease family, the integrity of the C-terminal di-glycine motif is essential for C-terminal binding, which must be placed in a shallow tunnel formed by two tryptophan residues and the simple substitution of glycine for alanine precludes the interaction[30,31], as probably occurs in SENP2 and SENP7. On the other hand, since the reaction between SUMO2 DHA precursor and USPL1 is highly efficient, we envision a different binding mechanism for the interaction of the C-terminal tail of SUMO2 with the active site groove of USPL1, compared to the SENP protease family.

**Overall structure of the USPL1-SUMO2 complex.** The USPL1-SUMO2 complex was formed by incubation of USPL1 with the SUMO2 DHA precursor at 37 °C for 2 h to form the covalent bond between the DHA group with the active site cysteine of USPL1 (Supplementary Fig. 2). A few diffraction quality crystals of the USPL1-SUMO2 complex were obtained in the initial screening in a condition containing 0.2 M potassium thiocyanate, 0.1 M sodium acetate pH 5.5, 8% w/v PEG20000, and 8% w/v PEG500MME, which was hard to reproduce in subsequent screenings. Molecular replacement with available USP models did not work due to the lower sequence identity with USPL1 (<15%), but fortunately, we were able to solve the structure by using the recently reported USPL1 model from the alpha-fold server[32], which unambiguously resulted in a correct final solution. The crystals contained one USPL1-SUMO2 complex per asymmetric unit and diffracted to a resolution of 1.8 Å. The final electron density map model of the USPL1 catalytic domain includes most of the residues (Ser225 to Ile501), with the only disruption of a disordered loop connecting two alpha helices in the thumb subdomain (Leu285 to Lys295).

USPL1 adopts the typical fold of the catalytic domain of the USP family, resembling the shape of a right-hand containing palm, fingers, and thumb subdomains (Fig. 2b and Supplementary Fig. 3). A Zn²⁺ atom coordinated by four cysteines stabilizes

the structure of the finger domain, as occurs in several USPs, forming a zinc-finger motif (ZnF). In addition to the general structural role of the ZnF motif, in USP21 it is also important for the binding of the distal ubiquitin in a linear diUb substrate[33]. SUMO2 is grabbed by the USPL1-like right hand in the complex, with the C-terminal tail of the SUMO2 precursor extended towards the catalytic triad of the USPL1 active site formed by Cys236, His456, and Asp472 (Fig. 2b, c). Structural comparison between USPL1 in complex with SUMO2 with the apo form of USPL1 from the AlphaFold model indicates a very good overlapping of both structures, with a main-chain rmsd (root mean square deviation) of 0.8 Å (Fig. 2c). The structural overlapping also revealed a similar location of residues forming the active site catalytic triad, with a 3.8 and 3.6 Å distance between the Cys236 Sγ and the His456 Nδ1; and 2.8 and 2.9 Å between His456 Nε2 and Asp472 Oδ1, in the unbound and SUMO2-bound USPL1, respectively (Fig. 2c). Such distances suggest that the USPL1 catalytic triad is already preformed and the protease might be active in the absence of the SUMO substrate, in contrast to the apo structures of other UPSs, such as USP7, USP15, and USP18, in which binding of the ubiquitin substrate is necessary to rearrange the catalytic triad to an active conformation[15,17,34].

**Structural comparison with ubiquitin-specific USP members.** As mentioned above, the sequence identity between USPL1 and the other USP members with deubiquitinase activity is only around 15%, which differs from the higher sequence identities shown among the ubiquitin-specific USP members (Fig. 3a and Supplementary Fig. 1). This might be a consequence of the deSUMOylase activity displayed by USPL1 in contrast to the deubiquitinase activity for all other USPs. Structural overlapping between USPL1 with USP7 (rmsd 3.02 Å, 197 aligned, 14.21% identity) and with USP28 (rmsd 2.61 Å, 184 aligned, 14.67% identity) are low, only showing a good superposition in the secondary structure elements encompassing the catalytic triad in the active site in the palm subdomain, namely the α1 helix (Cys236) and the β9 and β10 strands (His456 and Asp472) (Fig. 3a, b). In general, the thumb and fingers subdomains of USPL1 show very little homology with USP7 and USP28, particularly in the different length and orientation of α3, α4, and α5 helices in the thumb subdomain, and in the beta sheet containing the ZnF motif in the fingers subdomain[15,35].

Interestingly, a major difference in USPL1 is the lack of two loops in the palm subdomain, namely so-called blocking loops,

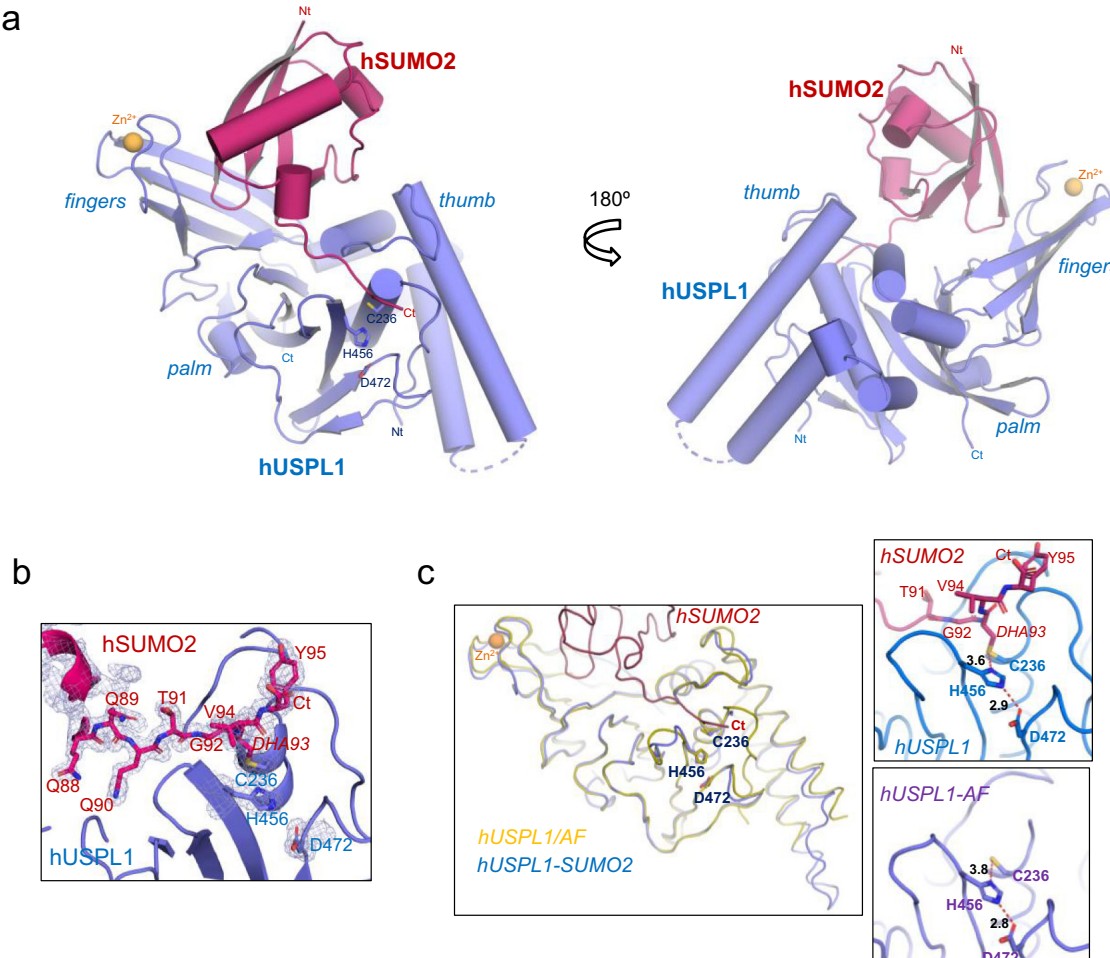

**Fig. 2 Crystal structure of the complex of human USPL1 with SUMO2 precursor. a** Two views of the USPL1-SUMO2 complex structure are shown in cartoon representation. USPL1 catalytic domain and SUMO2 precursor are shown in purple and red, respectively. USP right hand-like domains are labeled. The catalytic residues are labeled and depicted in stick representation. Zinc atom is shown in yellow. **b** Detailed view of the 2Fo-Fc electron density map contoured at 1σ of the C-terminal tail of SUMO2 bound to USPL1. Same color code as in (**a**). **c** Structural superimposition between USPL1-SUMO2 (blue) complex and the USPL1 AlphaFold (AF) model (yellow). Close-up views of the superimposition of active sites of USPL1 and AF model are shown in the right panel. Catalytic triad residues from USPL1 are labeled and shown in stick representation, whereas the corresponding residues from AF model are shown below. Hydrogen bonds are represented by dashed lines and the distances (Å) are also depicted.

between strands β6 and β7, and between β8 and β9 next to the catalytic triad (Fig. 3a–d). These two loops are highly conserved in the USP family due to their major role in ubiquitin binding (Supplementary Figs. 1 and 6) and they are structurally rearranged upon ubiquitin binding to trap ubiquitin within the right hand-like structure[15]. However, in USPL1 both loops are not required for binding to SUMO2. In fact, the superposition of USPL1 with USP28 displays a ~15° angle rotation of SUMO with respect to ubiquitin, which would result in a collision with the larger Blocking Loop (Fig. 3e). Thus, in addtion to the unique contacts with fingers and thumb subdomains, the lack of the blocking loops in USPL1 contributes to the different position of SUMO on the surface of USPL1 compared to ubiquitin, which is particularly evident in the different orientation of C-terminal tail backbone (Fig. 3f), and in the comparison of the electrostatic potential surfaces between USPL1 and USP28 (Fig. 3g).

**SUMO2 interface of the C-terminal tail with USPL1.** As expected by the divergent C-terminal sequences of SUMO (-FQQQTGG) and ubiquitin (-VLRLRGG), the contacts engaged by the SUMO C-terminal tail with the active site cavity of USPL1 constitute key signatures for the specificity of USPL1 for SUMO2.

The electron density maps clearly show the covalent bond formed between DHA93 in SUMO2 and Cys236 in USPL1 (Fig. 2b), in which the sidechain and not the C-terminal carboxylate is crosslinked to Cys236 Sγ. This fact constrains the geometry of this region, as observed by the *cis* configuration of the Gly92-DHA93 peptide bond in the USPL1 complex (Fig. 4), in which strong hydrogen bonds between the nitrogen and carbonyl oxygen of the Gly92-DHA93 peptide bond are established with the USPL1 backbone (each 2.9 Å distance) (Fig. 4).

The extended conformation of the SUMO2 tail (Gln90-Thr91-Gly92) is comparable to ubiquitin, but the specific backbone hydrogen bonds are different. Whereas Gly92 engages similar hydrogen bonds with USPL1 as in the ubiquitin-USP28 complex, Thr91 forms two hydrogen bonds with the side chain of Tyr451 Oζ (3.2 and 2.7 Å distance for the backbone N and O, respectively) (Fig. 4). In ubiquitin USPs Tyr451 is occupied by a highly conserved histidine (Supplementary Fig. 4), forming a similar hydrogen bond with the backbone oxygen of ubiquitin Arg74, but in this case, the backbone nitrogen interacts with the β8-β9 blocking loop, absent in USPL1 (Supplementary Fig. 4).

The backbone oxygen of Gln90 forms a hydrogen bond with a water molecule that is fixed by contacts with His493 and Ser332,

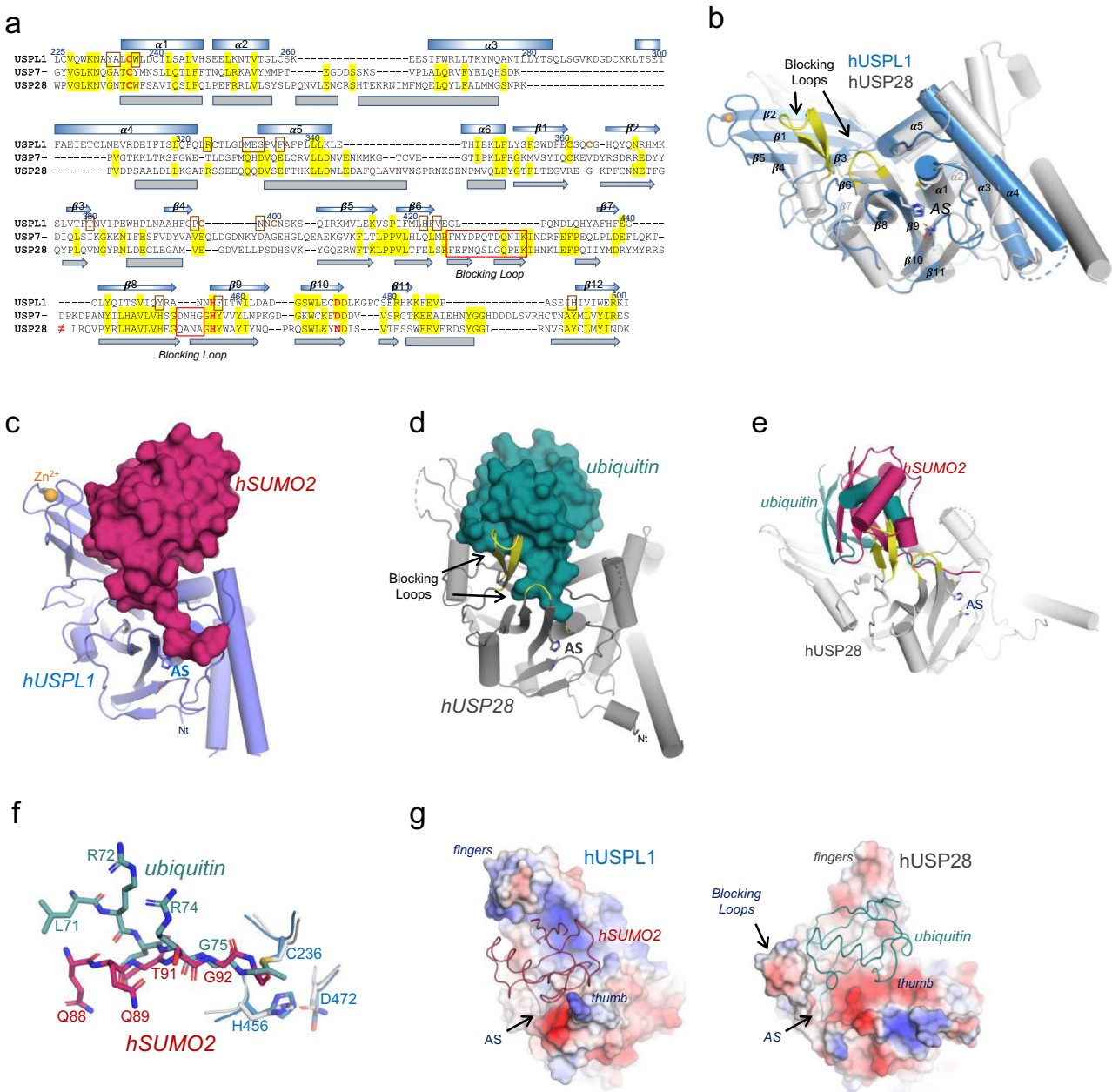

**Fig. 3 Comparative analysis of USPL1 with the ubiquitin USP28. a** Alignment of sequences corresponding to the catalytic domains for human USPL1, USP7, and USP28 based on the structural alignment of human USP7 (PDB code 4WPI) and USP28 (PDB code 6HEJ)[35,44]. Secondary structure elements are numbered (β-strands and α-helices) and indicated above the alignment for USPL1 (blue) and below the alignment for USP28 (gray). Gaps are denoted by --- and the large sequence insertion within USP28 is depicted ≠ to indicate that the sequence is missing from the alignment. Side chain identity (100% conservation) is denoted in the alignment by a yellow background. Three catalytic residues are depicted in red. Active site-blocking loops are denoted by red squares. All images were prepared with PyMOL[45]. **b** Cartoon representation of the structural overlapping of the catalytic domains of USPL1 with USP28 (PDB code 6HEK)[35]. The blocking loops connect β-strand 6 and β-strand 7, and β-strand 8 and β-strand 9 of the palm domain are missing in USPL1 and shown in yellow. Active site residues (AS) are shown in stick representation. Zinc atom is shown in yellow. **c** Surface representation of SUMO2 (red) in complex with USPL1, shown in cartoon representation (blue). Active site residues (AS) are shown in stick representation. Zinc atom is shown in yellow. **d** Surface representation of ubiquitin (dark green) in complex with USP28 (PDB code 6HEK)[35], shown in cartoon representation (gray). Active site residues (AS) are shown in stick representation. Zinc atom is shown in yellow. Blocking loops (yellow) are indicated. **e** Structural comparison of C-terminal tails of SUMO2 (red) and ubiquitin (green) in complex USPL1 and USP28, respectively. **f** Structural overlapping of SUMO2-USPL1 and ubiquitin-USP28 displaying the collision of the blocking loop (yellow) of USP28 (gray) with SUMO2 (red). **g** Electrostatic potential surface representation for USPL1-SUMO2 and USP28-ubiquitin to highlight the differences between USPL2 and USP28 in the analogous surface. SUMO2 (red) and ubiquitin (dark green) are shown in a line representation.

both highly conserved in USPL1 (Fig. 4 and Supplementary Fig. 1). In ubiquitin the backbone oxygen of Leu73 (equivalent to Gln90 in USPL1) forms a hydrogen bond with the side chain of a tyrosine (replaced by Phe457 in USPL1) (Supplementary Fig. 4).

Such tyrosine, located one position after the active site histidine, is highly conserved in all ubiquitin-specific USPs (Fig. 3 and Supplementary Fig. 1), but has been substituted by phenylalanine in all USPL1 orthologs, which forms hydrophobic contacts with

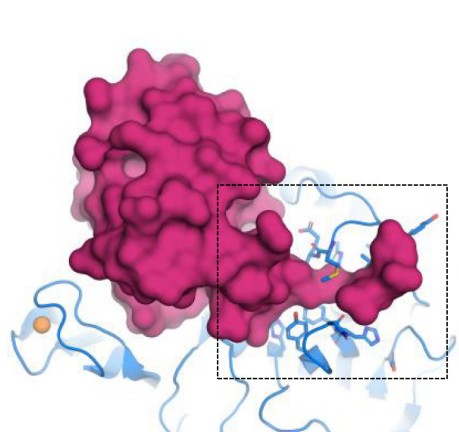

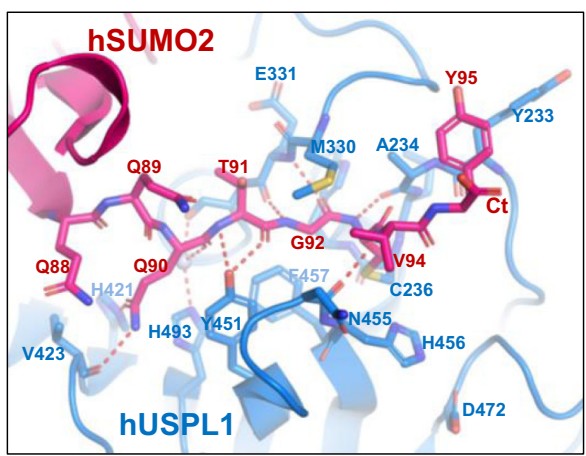

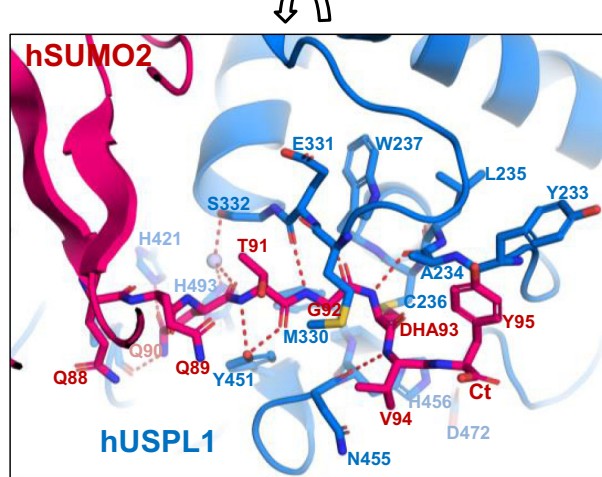

**Fig. 4 Atomic details of the C-terminal tail interaction of SUMO2 with USPL1.** Two views of the stick representation of the main contacts of the C-terminal of SUMO2 in complex with the active site groove of USPL1. USPL1 catalytic domain and SUMO2 are shown in blue and red, respectively. SUMO2 and USPL1 interface residues are labeled. Dashed lines indicate hydrogen bond contacts. On the left side, surface representation of SUMO2 (red) in complex with USPL1, shown in a blue cartoon. Interface residues are shown in stick representation. Zinc atom is shown in yellow.

Gly92 but is unable to establish a hydrogen bond as in ubiquitin. In addition, the side chain of Gln90 forms a hydrogen bond with a conserved His421 (3.0 Å distance) and with the backbone oxygen of Val423. Finally, the C-terminal tail of SUMO2 (Gly92) is sandwiched between Phe457 and Met330, substituted by a highly conserved glutamine in ubiquitin-specific USPs.

A relevant difference compared to the ubiquitin-specific USPs complexes is the extensive interface established by the two arginines and the two leucines in the ubiquitin tail (Leu71-Arg72-Leu73-Arg74), which participate in electrostatic and hydrophobic contacts on opposite sides of the ubiquitin tail. Particularly relevant is the interaction of the two ubiquitin leucines with the blocking loops, absent in USPL1, and the ubiquitin Arg72 and Arg74 with Glu258 and Gln254 in the ubiquitin-USP28 complex (Supplementary Fig. 4) (PDB:6HEK)[35]. In contrast, in USPL1 only Gln90 from the equivalent Gln88-Gln89-Gln90-Thr91 tail is engaged in a specific interaction with His421, indicating that this interface might be less relevant in USPL1 for SUMO binding.

The complex with USPL1 was formed with the SUMO2 precursor, which contains the complete or immature C-terminal tail formed by Val-Tyr extension after the proteolytic cleavage site (di-Glycine motif). Both residues at the SUMO2 C-terminal tail are well observed in the electron density maps, but probably their structural conformation is constrained by the covalent crosslink between DHA93 and the USPL1 Cys236.

**SUMO2 Interface with the thumb and fingers subdomains**. Two major-specific contacts stand out in the interaction between the USPL1 thumb subdomain and the SUMO2 surface, namely Arg324 and Phe335. Arg324 is engaged in a strong well-oriented electrostatic interaction with Asp71 of SUMO2, with 2.7 and 3.1 Å distances between the NH1 and NH2 of the Arg324 guanidinium group and the OD1 and OD2 of the Asp71 carboxylate group, respectively (Fig. 5a). Arg324 is well conserved in all USPL1 orthologs and this interaction is unique to USPL1, not present in ubiquitin-specific USPs. Interestingly, SUMO2 Asn68 and Asp72 were important determinants for the specificity of SENP7 for SUMO2/3 over SUMO1[36,37], where they are substituted by alanine and histidine, respectively. In USPL1, the proteolytic activity against SUMO1 is low compared to SUMO2 (Fig. 6), however, removal of the side chain of Arg324 did not increase the activity for SUMO1 substrates, as observed in activity assays with the USPL1 R324A point mutant (Supplementary Fig. 5).

Phe335 is buried in a hydrophobic groove in the SUMO2 surface formed by Pro66, Phe87, and the aliphatic chains of Arg59 and Arg61 (Fig. 5a). Phe335 is highly conserved in all USPL1 orthologs and not observed in other USPs. In ubiquitin-specific USPs this location is occupied by a conserved glutamate (Glu258 in USP28), which is engaged in a strong electrostatic bridge interaction with Arg72, from the ubiquitin

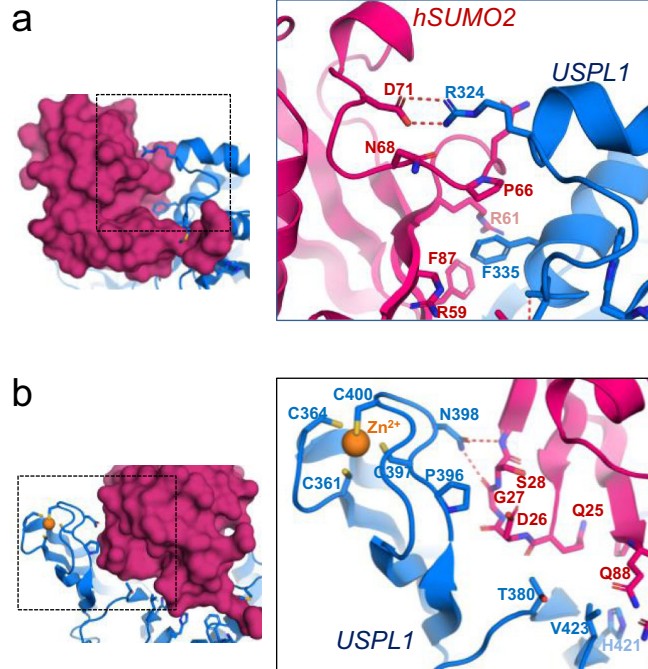

**Fig. 5 SUMO2 interface with the thumb and fingers subdomains of USPL1. a** Stick representation of the main contacts of the thumb subdomain of USPL1 with SUMO2. USPL1 catalytic domain and SUMO2 are shown in blue and red, respectively. SUMO2 and USPL1 interface residues are labeled. Dashed lines indicate hydrogen bond contacts. On the left side, surface representation of SUMO2 (red) in complex with USPL1, shown in a blue cartoon. Interface residues are shown in stick representation. Zinc atom is shown in yellow. **b** Stick representation of the main contacts of the fingers subdomain of USPL1 with SUMO2. Same color code as in (**a**).

| | USPL1-SUMO2 |
|---|---|
| *Data collection* | |
| Space group | P2₁ |
| Unit cell parameters (Å) | 50.71, 69.88, 53.64 |
| Wavelength (nm) | 0.97625 |
| Resolution range (Å) | 41.04–1.79 |
| $R_{merge}$ | 0.08 (0.47)[a] |
| $R_{pim}$ | 0.06 (0.36)[a] |
| $(I/\sigma(I))$ | 7.8 (2.1)[a] |
| Completeness (%) | 96.5 (97.8)[a] |
| Multiplicity | 2.5 (2.5)[a] |
| CC (1/2) | 0.99 (0.72)[a] |
| *Structure refinement* | |
| Resolution range (Å) | 41.04–1.80 |
| No. of unique reflections | 33540 |
| $R_{work}/R_{free}$ (%) | 17.6/19.7 |
| No. of atoms | |
| Protein | 2823 |
| Water molecules | 200 |
| Zn²⁺ | 1 |
| Overall B factors (Å²) | 33.99 |
| USPL1 (Å²) | 31.10 |
| SUMO2 (Å²) | 42.66 |
| Zn²⁺ (Å²) | 24.22 |
| Water molecules (Å²) | 38.29 |
| Rms deviations | |
| Bonds (A) | 0.007 |
| Angles (°) | 0.845 |
| Ramachandran favored (%) | 97.94 |
| Ramachandran allowed (%) | 1.77 |
| Ramachandran outliers (%) | 0.29 |
| *PDB code* | 7P99 |

**Table 1 Crystallographic statistics of the USPL1-SUMO2 complex.**

[a]Data from the last shell in parenthesis (1.70–1.79 Å).

C-terminal tail (Supplementary Fig. 4). Both Phe335 in USPL1 and Glu258 in USP28 probably have an important impact on the binding and specificity of SUMO2 or ubiquitin, respectively.

In the USPL1 fingers subdomain, a Zn²⁺ atom stabilizes the structure by the coordination of four conserved cysteine residues (Fig. 5b), and as observed in other USPs, the cysteine coordination to Zn²⁺ is essential for the USP catalytic activity. In USPL1, the contacts are basically engaged by the β1-β2 hairpin loop of SUMO2, which nicely fits in the fingers subdomain surface. The only specific side-chain contact is engaged by Asn398, located next to the Zn²⁺ site and conserved in all USPL1 orthologs, which forms two hydrogen bonds with the main chain oxygen and nitrogen of Gly27 and Ser28 in SUMO2, respectively (2.8 and 3.0 Å distances). In ubiquitin-USP complex structures, the fingers subdomain normally display a rigid body adjustment to fix ubiquitin, sometimes having an impact in chain specificity, as occurs in USP21[33]. However, in the USPL1-SUMO2 complex, the orientation of the fingers subdomain is similar to the AlphaFold-2 model of the apo form (Fig. 2) and displays a different orientation compared to all known structures of ubiquitin USPs, showing an average 14 Å displacement between the Zn²⁺ atoms of the ZnF motifs (Supplementary Fig. 6). As a consequence, the contacts observed at the interface of the fingers subdomain with SUMO2 are not observed in the other USP-ubiquitin complexes, basically due to the different orientation of the fingers subdomain with respect to SUMO2.

**Mutagenesis analysis of the specific interface contacts in USPL1.** USPL1 shows a proteolytic isoform preference for human SUMO2, in contrast to other SUMO peptidases such as SENP2

which shows similar activities for SUMO1 and SUMO2-AMC substrates (Fig. 6a)[31]. As expected, USPL1 does not show any activity against ubiquitin-AMC substrate (Fig. 6a)[18]. To characterize the binding interface in this kinetically trapped intermediate complex, specific interactions in the USPL1-SUMO2 interface have been mutated. The overall structural integrity of all USPL1 mutants is similar to the wild-type form, as observed by comparing their purification profiles by ion-exchange chromatography and by the similar spectra displayed in the intrinsic Trp-fluorescence emission analysis (Supplementary Fig. 7). The USPL1 mutant interface analysis has been conducted in the C-terminal tail, thumb, and fingers subdomains (Fig. 6c). Binding to SUMO2 has been checked by using a SUMO2 DHA chemical trap substrate, and the proteolytic activity has been checked against the SUMO2-AMC fluorescent substrate, as well as by RanGAP-SUMO2 and diSUMO2 substrates (Table 1).

In the C-terminal tail interface, Phe457 has been replaced either by tyrosine, which is present in all USP members, or by leucine, which maintains the hydrophobic character of the interaction. In both F457Y and F457L mutants the binding reaction with SUMO2 DHA is strongly diminished and the catalytic activity against all tested substrates is reduced (Fig. 6b–f). Interestingly, despite that all other USP members contain a tyrosine in that position, the only presence of a Oζ from the tyrosine side chain seriously compromises the interaction of USPL1 with SUMO2 substrates. The SUMO C-terminus is sandwiched between Phe457 and Met330 (Fig. 6c), however, removal of Met330 side chain has a minor effect on the catalytic activity and even seems to increase the binding affinity for SUMO2, as observed in the M330A mutant (Fig. 6b).

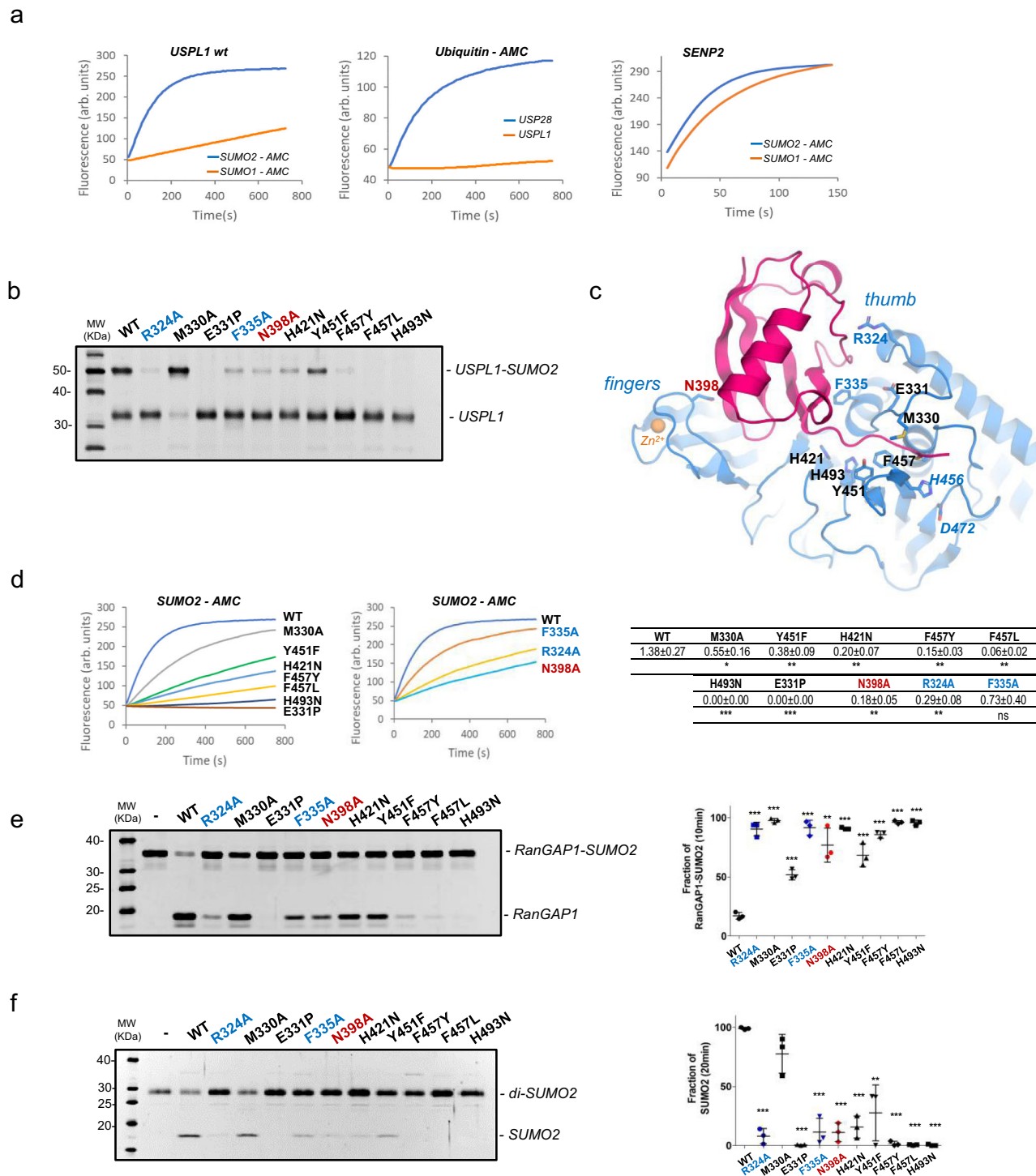

While Met330, which is normally substituted by glutamine in ubiquitin USPs, may still be involved in regulating USPL1 activity, it does not seem to be essential for binding and catalysis. However, the backbone interactions of the adjacent Glu331 with the SUMO C-terminal tail are essential for binding and catalysis, as observed in the E331P mutant, in which a proline substitution distorts the backbone orientation and removes a critical hydrogen bond (Fig. 6b–f).

Removal of the Oζ from the tyrosine side chain in the Y451F mutant decreases binding and activity by fourfold compared to wild-type (Fig. 6b–f), highlighting the role of the two hydrogen bonds with Thr91 of the C-terminal tail of SUMO2 (Fig. 4).

Finally, two conserved histidine residues were analyzed: H421N removes the specific hydrogen bond contact with Gln90, and H493N removes the interaction to a fixed water (Fig. 4). Interestingly, whereas the H421N mutant affects partially binding and catalysis, the integrity of the His493 side chain, which bridges the C-terminal SUMO backbone through a water molecule, seems essential for binding and catalysis. The overall structural integrity of the H493N mutant seems correct, as observed in Trp-fluorescence stability analysis compared to wild-type (Supplementary Fig. 7), thus highlighting the relevance of this water-bridged interaction in the overall hydrogen bond network of the C-terminal tail.

**Fig. 6 Functional analysis of the SUMO2-USPL1 interface. a** Left, activity assay of USPL1 with SUMO1-AMC and SUMO2-AMC. Middle, activity assay of Ubiquitin-AMC with USPL1 and USP28. Right, activity assay of SENP2 with SUMO1-AMC and SUMO2-AMC. Reactions were conducted in triplicate and the average curve is displayed. **b** Binding interaction of USPL1 point mutants with SUMO2-DHA. Reaction assays were conducted with USPL1 wild-type and mutants for 3 h. $n = 3$ technical replicates. **c** USPL1-SUMO2 complex structure shown in cartoon representation. USPL1 catalytic domain and SUMO2 precursor are shown in purple and red, respectively. USP right hand-like domains are labeled. Interface residues are labeled and shown in stick representation. Zinc atom is shown in yellow. **d** Left, activity assays of USPL1 wild type and mutants of the C-terminal tail interface using SUMO2-AMC. Middle, similar activity assays of USPL1 wild type and mutants of the thumb and fingers subdomains. Right, table indicating the mean slope values plus/ minus the standard deviation of the activity assays. $n = 3$ technical replicates. Significance was measured by a two-tailed unpaired $t$-test relative to wild-type. *$P < 0.05$, **$P < 0.01$, ***$P < 0.001$. **e** Left, endpoint assays of USPL1 wild type and mutants using the RanGAP1-SUMO2 substrate. Right, plot of the RanGAP1-SUMO2 fraction after 10 min reaction. Data values represent the mean ± SD, $n = 3$ technical replicates. Significance was measured by a two-tailed unpaired $t$-test relative to wild-type. *$P < 0.05$, **$P < 0.01$, ***$P < 0.001$. Exact $P$ values from left to right: <0.0001, 0.0868, <0.0001, 0.0002, <0.0001, 0.0001, 0.0065, <0.0001, <0.0001, <0.0001. **f** Left, endpoint assays of USPL1 wild type and mutants using di-SUMO2 substrate. Right, plot representation of the product SUMO2 fraction after 20 min reaction. Data values represent the mean ± SD, $n = 3$ technical replicates. Significance was measured by a two-tailed unpaired $t$-test relative to wild-type. *$P < 0.05$, **$P < 0.01$, ***$P < 0.001$. Exact $P$ values from left to right: <0.0001, <0.0001, 0.0003, <0.0001, 0.0021, <0.0001, 0.0009, <0.0001, <0.0001, <0.0001. Source data for Fig. 6a, b, d, e, and f are provided as a Source Data file.

Two specific contacts to SUMO2 have been analyzed in the USPL1 thumb subdomain, Arg324, and Phe335, both highly conserved in the USPL1 orthologs (Supplementary Fig. 1). The R342A mutant removes a strong and well-oriented electrostatic bridge with Asp71 of SUMO2 (Fig. 5a), and the results indicate that binding and catalysis with SUMO2 substrates are seriously compromised, highlighting a major role for Arg324 (Fig. 6b–f). Likewise, in a lesser degree compared to Arg324, the F335A mutant also disturbs binding and catalysis for SUMO2 substrates. Thus our in vitro proteolytic activities as well as the strong conservation in the family indicate that both Arg324 and Phe335 play a major role in the specific interaction of SUMO2 with USPL1.

The structural architecture of the fingers subdomain is maintained by four conserved cysteine coordination residues forming a ZnF motif (Fig. 5b). Despite the extended interface of the fingers subdomain with SUMO, only the highly conserved Asn398 seems to establish specific contacts by the formation of two hydrogen bonds with SUMO2 backbone atoms (Fig. 5b). Interestingly, removal of these interactions in the N398A mutant seriously compromises binding and catalysis with SUMO2 substrates. Altogether, the specific contacts of the thumb and fingers subdomain, Arg324, Phe335, and Asn398, have an essential contribution to fix the globular domain of SUMO2 in a correct orientation for catalysis.

## Discussion

All members of the human USP family of DUBs show specificity towards ubiquitinated substrates, with the exception of USP18, which is specific towards ISG15, a double-headed UbL modifier similar to ubiquitin, and USPL1, which shows specificity towards SUMO, a different UbL modifier. USPs are multidomain proteins that utilize different strategies to ensure their specific functions in the cell. In some cases, the presence of additional domains adjacent to the USP catalytic domain enhances substrate binding or provides specificity towards a particular type of polyubiquitin chain. Commonly, in all members of the USP family, the structural fold of the catalytic domain, formed by the characteristic palm, thumb, and fingers subdomains reminiscent of a human right hand, is maintained, exposing all the key surface elements necessary for interaction with ubiquitin. However, as described here, USPL1 is a distant member of the family with the canonical right-hand scaffold of a ubiquitin USP catalytic domain, but exposing key surface elements unique for SUMO binding, a low-identity UbL in comparison to ubiquitin.

Possibly, a major difference between USPL1 and the other USP members can be found in the interface formed by the dissimilar C-terminal tail of ubiquitin (LRLRGG) and SUMO (QQQTGG) (Fig. 7). In addition to different hydrogen bond contacts engaged

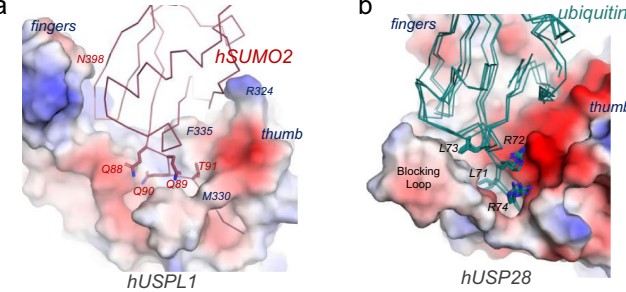

**Fig. 7 Comparison of the C-terminal tails of SUMO2 and ubiquitin. a** Electrostatic potential surface representation for the USPL1 in complex with SUMO2 (red line). The C-terminal tail of SUMO (QQQTGG) is labeled and shown in stick representation. Main interface contacts of thumb and fingers subdomains are labeled. **b** Electrostatic potential surface representation for the USP28 (PDB code 6HEK) in complex with ubiquitin (green lines) from complexes with USP2 (PDB code 2hd5), USP7 (PDB code 5JTV), and USP30 (PDB code 5OHK)[35,46-48]. The C-terminal tail of ubiquitin (LRLRGG) is labeled and shown in stick representation.

by the backbone, the presence of two arginines and two leucines in opposite sides of the ubiquitin tail generates unique interactions with acidic and hydrophobic surface patches in all ubiquitin USPs (Fig. 7), including USP18, in which ISG15 has a similar tail as ubiquitin. Interestingly, such hydrophobic patch in all ubiquitin-specific USPs is formed by the so-called blocking loops, which are structurally rearranged upon binding to the ubiquitin substrate. Interestingly, in all USPL1 orthologs, the two blocking loops sequences have been deleted and are absent in the structure. Thus, in contrast to the high number of specific contacts engaged by the -LRLR- motif of the ubiquitin C-terminal tail, in the USPL1-SUMO2 structure only the Gln90 side chain of the equivalent -QQQT- motif forms specific interactions with USPL1, and all other contacts in the C-terminal tail of SUMO are established by backbone atoms (Fig. 4).

It is interesting to note how USPL1 has evolved to interact with SUMO. Key USP major residues for specific interaction with ubiquitin have been deleted or, in some cases, replaced by residues specific for interacting with SUMO2 (Fig. 3 and Supplementary Fig. 1). The di-glycine motif in the C-terminal tail of SUMO, also conserved in ubiquitin, is sandwiched by a narrow cleft formed by Phe457 and Met330, both located on opposite sides of Gly92. Whilst Phe457 is highly conserved in all USPL1 orthologs, it is replaced by tyrosine in all ubiquitin USPs, which forms a hydrogen bond with the ubiquitin C-terminal backbone[15]. Interestingly, a phenylalanine substitution by

tyrosine in USPL1, F457Y, which mimics ubiquitin USPs, shows decreased binding and reduced catalysis of SUMO2 substrates (Fig. 6), underscoring the relevance of single contacts at the USPL1-SUMO2 interface. Another remarkable difference is Tyr451, conserved in all USPL1 orthologs and replaced by histidine in all ubiquitin USPs (Figs. 3a, 4 and Supplementary Fig. 1), and both forming hydrogen bonds with the C-terminal tail backbone. While removal of this hydrogen bond interaction is crucial in ubiquitin USPs[15], binding and catalysis of SUMO2 substrates with the Y451F mutant on USPL1 is reduced by approximately fourfold compared to wild-type (Fig. 6). Both residues, F457 and Y451, are partly responsible for the different hydrogen bond networks established in the C-terminal tails of SUMO and ubiquitin.

Probably in USPL1, an ultimate adaptation to bind SUMO2 has occurred through specific changes in the fingers and thumb subdomains. Arg324 and Phe335 in the thumb subdomain, and Asn398 in the fingers subdomain, are engaged in specific interactions with SUMO2 residues. Interestingly, all three residues are highly conserved in all USPL1 orthologs and their substitution by alanine seriously compromises the binding and catalytic activity of SUMO2 substrates (Fig. 6). These residues have no equivalent in any ubiquitin-specific USPs, basically, as a consequence of the low sequence identity in those subdomains (Fig. 3), resulting in an ~10–15° rotation of the SUMO globular domain with respect to ubiquitin in the complex. As a consequence, the SUMO interface with the fingers and thumb subdomains does not share any similar interaction compared to the other ubiquitin-specific USP complexes, which involve different contacts at these subdomains.

In this manuscript, we want to emphasize the plasticity of the conserved catalytic domain of USPs, which despite being designed to bind ubiquitin with high specificity, has evolved to interact with SUMO in USPL1. SUMO and ubiquitin share a 16% sequence identity, basically observed in the structural elements of the globular domain. However, the non-conserved residues in their surface participate in unique and specific protein–protein interactions with USP family members. USPL1 is a paradigmatic example of divergent evolution in the USP family, in which the aforementioned surface substitutions facilitate the interaction with SUMO but maintain the right hand-like subdomain scaffold of the catalytic domain.

It is also interesting to compare the SUMO interface between USPL1 and members of the SUMO-specific SENP/ULP protease family. Interestingly, in this family, all members are specific for SUMO except SENP8/DEN1, which has proteolytic activity for Nedd8-conjugated substrates[21–24]. The total interface with SUMO2 is quite large in both cases, 1169 and 1070 Å$^2$ for USPL1 and SENP2 (PDB 2IO0)[38], respectively. However, excluding the particular contacts with the extended C-terminal tail of SUMO2 in each family, almost no similarities are observed at the interface with the globular domain of SUMO. As an example, in a similar interface region in both USPL1 and SENP2, Arg59, Arg61, and Phe87 are engaged in salt bridge and hydrophobic contacts with SENP2[38]. However, in USPL1, those salt bridges are not present, but the two arginines, together with Phe87, form a hydrophobic pocket that interacts with Phe330 on USPL1, which is essential for binding and inhibition (Figs. 5a and 6).

The catalytic domain of USPL1 is embedded in the middle of a structurally disordered full-length protein, which does not appear to have any other obvious well-folded domain in the sequence, as seen in the predicted model from Alphafold-2[32]. This type of domain organization of the full-length protein is often present in the SENP/ULP protease family, in which long-disordered protein extensions are usually found adjacent to the catalytic domain[10]. As for the SENP/ULP family, we cannot rule out a functional role for these long non-catalytic disordered extensions of the full-length USPL1, such as in the recruitment of SUMO substrates or in the regulation of the proteolytic activity of the catalytic domain, perhaps by internal protein–protein interactions. USPL1 has been reported to be involved in the regulation of the RNA polymerase-II-mediated snRNA transcription in the Cajal Bodies (CBs), a membrane-less compartment in the nucleus where USPL1 has been localized. However, the essential function of USPL1 in the CBs biology seems to be independent of the SUMO protease activity, perhaps the long-disordered extensions of USPL1 participate in protein–protein interactions of this membrane-less compartment. In any case, the relevance of the SUMO protease activity of USPL1 in the cellular context still needs to be disclosed.

In summary, we have unraveled the structural determinants for the unique specificity of USPL1 for SUMO2, a paradigmatic example of divergent evolution in the ubiquitin USP family to interact with a distant UbL family member.

## Methods

**Plasmids, cloning and point mutation**. HA-USPL1-pcDNA3.1 was a gift from Frauke Melchior (Addgene plasmid #85760; http://n2t.net/addgene:85760; RRID:Addgene_85760)[18]. The catalytic domain construct pET28a-USPL1CD was amplified by PCR using Phusion polymerase and cloned into the BamHI/NotI restriction enzymes sites of pET28a vector using ligation. The USPL1 point mutants constructs were designed by different primers and were created by the QuickChange site-directed mutagenesis kit (Stratagene). All primers are shown in Supplementary Table 1. pET28a-Δ14-human SUMO2 (14 amino acids deletion) was constructed at the Sloan-Kettering Institute in New York by David Reverter. The plasmid of pET28a-Δ14-SUMO2GCVY(C48S) has been generated by Restriction Enzyme Free PCR[39].

**Protein expression and purification**. The USPL1 CD and Δ14SUMO2 expression constructs were transformed into *E. coli* Rosetta (DE3) cells (Novagen). Bacteria were grown at 37 °C to OD600 = 0.7~0.8, and IPTG was added to a final concentration of 0.5 mM. Bacteria were grown an additional 16 h at 20 °C and harvested by centrifugation. Cell suspensions were equilibrated in 350 mM NaCl, 20 mM Tris-HCl (pH 8.0), 10 mM imidazole, 20% sucrose, 1 mM DTT, and 0.1% IGEPAL CA-630, and cells were broken by sonication. After removing cell debris by centrifugation, proteins were separated from lysate by nickel affinity chromatography using Ni Sepharose 6 Fast Flow (GE Healthcare) and eluted with lysis buffer including 20 mM Tris-HCl (pH 8.0), 350 mM NaCl, 300 mM imidazole, and 1 mM DTT. Fractions containing the target protein were collected, diluted to 50 mM NaCl, applied to an anion exchange resin (Resource Q; GE Healthcare), and eluted with a 0–1 M NaCl gradient from 0 to 35% in 20 mM Tris-HCl (pH 8.0) and 3 mM DTT. Concentrated the protein using Amicon Ultra-30K ultrafiltration device (Millipore) and snap-frozen in liquid nitrogen prior to storage at −80 °C.

**Preparation of the USPL1CD- Δ14-SUMO2 DHA complex**. After purification of Δ14-SUMO2GCVY (C48S) protein, 2 mM DTT was added as a solid to a 500 μl aliquot of protein solution to reduce any contaminant disulfide and gently shaking 15 min. The buffer was changed to 20 mM Tris8, 150 mM NaCl by PD-10 (GE Healthcare) and concentrated to 10 mg/ml and kept at room temperature for the next step. A stock solution of 2,5-dibromo hexanediamide (DHA) was prepared by dissolving 35.5 mg in 418μl DMF. A 10-fold molar DHA (25 μmol) was added into Δ14-SUMO2GCVY (2.5 μmol). The reaction mixture was incubated at room temperature for 30 min and then at 37 °C for 10 h. The insoluble dibromide was removed using centrifugation and further purified using a Resource Q column to get Δ14-Sumo2 DHA protein. The mixture of USPL1CD and Δ14-Sumo2 DHA (1:3 molar ratio) was incubated at 30 °C for 3 h. Anion exchange chromatography (Resource Q; GE Healthcare) and gel filtration chromatography (Superdex 75; GE Healthcare) were carried out to purify the USPL1CD-Δ14SUMO2DHA complex.

**Crystallization and data collection**. The complex USPL1CD-Δ14SUMO2DHA was concentrated to 8 mg/mL for crystallization screening in a buffer containing 20 mM Tris 8.0, 170 mM NaCl and 1 mM DTT. Crystallization experiments were performed at 18 °C by the sitting drop vapor diffusion method and crystals grew up in a protein mixture with an equal volume of a condition solution containing 0.2 M potassium thiocyanate, 0.1 M sodium acetate 5.5, 8% w/v PEG20000, and 8% w/v PEG500MME. Crystals were harvested after 3 days and soaked 5–10 s in the crystallization buffer supplemented with 15% ethylene glycol, and then snap-frozen in liquid nitrogen to storage.

Diffraction data were collected to 1.8 Å resolution at beamline ID30B at the ESRF (Grenoble, France). Data processing was conducted by AutoProcessing with MxCUBE[40,41]. The space group was P2$_1$ and there was one complex per

asymmetric unit. The structure of USPL1CD-Δ14SUMO2DHA was solved by molecular replacement with USPL1 Alphafold2 model as a search mode[32]. Following rounds of model building and refinement were carried out with Coot and Phenix[42,43]. The structure of USPL1CD-SUMO2 has been deposited in the Protein Data Bank under accession codes PDB 7P99.

**SUMO-AMC hydrolysis assays**. USPL1 wild type and mutants were incubated with ubiquitin-, SUMO1- or SUMO2-AMC at 30 °C and measured the fluorescence emission using 345 nm excitation and 445 nm emission wavelengths using a Jasco FP-8200 spectrofluorometer. All measurements were carried out in triplicate with 1 nM USPL1 and 0.1 μM SUMO-AMC in a buffer containing 100 mM NaCl, 20 mM Tris-HCl pH 8, 10 mM DTT.

**In vitro de-SUMOyation assays**. Protease activity was measured by incubating di-SUMO2 and NΔ419RanGAP1-SUMO2 (1 μm) with purified 20 nM of USPL1 wt and mutants at 37 °C in a buffer containing 40 mM Tris-HCl (pH 8.0), 250 mM NaCl, and 2 mM DTT. Reactions were stopped after 0, 5, 10, and 20 min with SDS-BME loading buffer and analyzed by gel electrophoresis (PAGE). Proteins were detected by staining with SYPRO Ruby protein gel stain (Invitrogen). Products were detected using a Gel-Doc XR system (BioRad) and quantified with ImageLab software (Bio-Rad). The fraction of analyzed bands were plotted as mean ± SD. Statistical analyses were performed with Graphpad Prism 5.0. Significances were measured by a two-tailed unpaired $t$-test relative to wild type. All data were analyzed with a 95% confidence interval.

**Intrinsic fluorescence measurements**. Intrinsic fluorescence spectra were recorded using a Jasco FP-8200 spectrofluorometer. Tryptophan emission spectra were obtained by setting the excitation wavelength at 295 nm and collecting emission in the 315–400 nm range. USPL1 (wild-type and mutants) were diluted to achieve 1 μM in 20 mM HEPES pH 7.5 buffer containing 250 mM NaCl and 10 mM DTT. The temperature was set at 30 °C. For thermostability at 30 °C proteins were kept at this temperature for up to 60 min and Trp spectra were recorded at different times.

**Reporting summary**. Further information on research design is available in the Nature Research Reporting Summary linked to this article.

## Data availability

The structure reported has been deposited in the Protein Data Bank under accession code 7P99. Other Protein Data Bank accession codes used in this study: 6HEK (USP28-ubiquitin); 2HD5 (USP2-ubiquitin); 5JTV (USP7-ubiquitin); 5OHK (USP30-ubiquitin). All other data supporting the findings of this study are available within the article and its supplementary information files. Source data are provided with this paper.

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

## Acknowledgements

This work was supported by grants from the *"Ministerio de Ciencia, Innovación y Universidades"* PGC2018-098423-B-I00 to D.R. Y.L. acknowledges her scholarship of the China Scholarship Council program from the Chinese government. D.R. acknowledges support from the Serra Hunter program from Generalitat de Catalunya. We acknowledge the European Synchrotron Radiation Facility for the provision of synchrotron radiation facilities, and we would like to thank Andrew McCarthy for assistance in using beamline ID30B.

## Author contributions

Y.L. conducted the crystallization experiments. Y.L. and N.V. conducted in vitro activity assays. Y.L., N.V. and D.R. contributed to the correction and writing of the manuscript.

## Competing interests

The authors declare no competing interests.
