## [Peer Review File · Nature Communications]

nature portfolio

Peer Review File

Draft OnlyREVIEWER COMMENTS

Reviewer #1 (Remarks to the Author):

Li et al. present the first structure of a distantly related USP family member, USPL1, which is specific for SUMOylated substrates rather than the more typical specificity for ubiquitinated substrates in this family. In addition, they present data to characterise the importance of individual residues for mediating this specificity. The manuscript contains very interesting novel data and adds to our understanding of the USP family and substrate specificity requirements and will thus advance the field. Before it can be considered for publication in Nature Communications the following points need however be addressed with regards to providing more clarity and detail.

In general, the (slight) overuse of the terms “unique” and “unusual” should be avoided as it is a distraction and careful proof-reading is required (please see for some of the points listed below). All crystal structures discussed and shown in the text and figures need to be properly referenced, not only the PDB codes quoted.

Line 17: the expression ‘proteolytic activity towards ubiquitin’ could be misinterpreted, it isn’t ubiquitin (mostly interpreted as mono-ubiquitin) that is proteolytically processed. Please be more precise.

Line 35: an isopeptide bond

Line 42: DUB families

Line 44: in humans

Line 65: Full-length USPL1...

Line 67 where it is...

Line 80: avoid term “ubiquitin USPs” for something more precise: USPs that deubiquitinate

Line 94: USP ubiquitin specific protease family:

Line 98: clarify: are USP7 and USP28 the closest related USP members with the highest shared sequence identity to USPL1 based on the catalytic domain? Clarify which USP has the highest shared sequence identity with USPL1.

Line 134: specify the crystallisation conditions in the text

Line 137: please replace “crystal phases” with a more accurate term.

Line 162 and elsewhere: replace “ubiquitin USP members” with “USP members acting on ubiquitinated substrates” or similar as the term ubiquitin USP is confusing (it equals: ubiquitin ubiquitin specific protease).

Line 173: "...the sequence identity is difficult to predict"; it shouldn't be as the stretches of sequence in these regions can still be aligned even if structures do not align well? Please be precise

Line 181 and elsewhere in the manuscript: the lack of the blocking loop.... Two active site loops are generally referred to as blocking Loop 1 and blocking Loop 2 in USPs, respectively. Please specify which of the two blocking loops is missing (or shorter?) in USPL1.

Line 199 and elsewhere: add reference and PDB code for where direct structural comparisons with published structures are described. Here: USP28-Ubiquitin complex structure.

Line 253: "a subtle rigid body adjustment to fix ubiquitin": please rephrase: in some USPs I wouldn't describe the conformational changes between free state and distal ubiquitin bound state in this area as subtle, these can reach several Å: for example, in USP12 the fingers domain opens up by over 8 Å. Also, for USP8 significant movement of the fingers region is anticipated to fit a ubiquitinated substrate. please revise.

Line 254: "the USPL1 the orientation of the fingers subdomain seems already fixed in the apo form": on which basis has this statement been made? There is no USPL1 structure in the free form available to justify this. Please revise.

"As a consequence, the fingers subdomain interface with SUMO2 is unique in USPL1 and do not share any contact with ubiquitin USPs complexes": please clarify the meaning of this. I suspect it is mainly about the observed orientation of the fingers domain with regards to the core of SUMO but this needs more detail for clarification.

Line 279, 280: the C-terminus... affinity for

Line 301: *in vitro* in italics

Line 377: rule out instead of discard

Table 1: needs more detail and be better annotated in places: what values exactly are quoted and what is shown in brackets. Show average B-factors for protease and SUMO-2 and waters separately. Include information on Ramachandran plot regions.

Method section: pET instead of PET; which buffer was the protein in for crystallisations, please add.

Line 710: which SYPRO stain specifically?

Discussion: the discussion is too narrow. What is missing are wider structural comparisons. Please include a paragraph on whether there are any shared characteristics with the SENP family of proteases. It is likely that there are some similarities despite the structural differences.

Figure 6: Error bars have to be shown for the time courses and the number or replicates or repeats stated in order to evaluate the statistical significance.

Suppl. Fig. 1b: On what basis have USP sequences shown be selected? This needs to be clearly stated.

Reviewer #2 (Remarks to the Author):

By sequence, USPL1 is member of the so-called USP class of Ubiquitin-specific proteases. Nonetheless, however, it does not act upon ubiquitin but instead is specific for SUMO2. The authors report a crystal structure of covalently trapped complex between the catalytic domain (residues 212-514) of human USPL1 and the human precursor of SUMO2 at 1.8 Å resolution.

Thereby, important differences in the interaction of USPL1 with SUMO2 versus ubiquitin-specific members of the USP family and ubiquitin are revealed.

It also shown that USPL1 has a strong preference towards SUMO2 over SUMO1 or ubiquitin.

The structural data identified a number of interactions that could explain the unusual preference of USPL1, the relevance for the action upon SUMO2 is supported by a mutational analysis for several of them including Arg 324, Phe335, Asn398, Tyr451, Phe457, and His493 in biochemical assays.

Altogether, this appealing structure/function analysis illuminates an interesting evolutionary diversification of specificities towards distinct Ubl family proteins starting from a conserved enzyme fold, that will be of interest to researchers in the field.

Specific comments on details:

There is a bit of a muddle with figure mentioning in the text:

Line 150: "Figure 1b,c" should probably be Figure 2b,c

Line 152: "Figure 1d" should probably be Figure 2d (but see below).

Figure 2D appears in the text (line 155) but neither in the figure nor its legend.

Line 164: Figure 2a, should probably be Figure 3a

It is concluded that M330 "does not seem critical for binding and catalysis." However, even though the M330A mutant appears to bind better in the assay presented on figure 6b, it shows diminished processing of RanGAP1-SUMO2 (Figure 6e) and also reduced reaction with SUMO2-AMC as compared to wt. This may not be "critical" but at least seems significant.

Line 288: One should not call it "conserved histidine mutants"

REVIEWER COMMENTS

Reviewer #1 (Remarks to the Author):

Li et al. present the first structure of a distantly related USP family member, USPL1, which is specific for SUMOylated substrates rather than the more typical specificity for ubiquitinated substrates in this family. In addition, they present data to characterise the importance of individual residues for mediating this specificity. The manuscript contains very interesting novel data and adds to our understanding of the USP family and substrate specificity requirements and will thus advance the field. Before it can be considered for publication in Nature Communications the following points need however be addressed with regards to providing more clarity and detail.

In general, the (slight) overuse of the terms “unique” and “unusual” should be avoided as it is a distraction and careful proof-reading is required (please see for some of the points listed below). All crystal structures discussed and shown in the text and figures need to be properly referenced, not only the PDB codes quoted.

R. Thanks for the advices. We have now restricted the use of those terms in the new version of the manuscript. We have also added the references to all the PDB structures mentioned in the main text and figures.

Line 17: the expression ‘proteolytic activity towards ubiquitin’ could be misinterpreted, it isn’t ubiquitin (mostly interpreted as mono-ubiquitin) that is proteolytically processed. Please be more precise.

R. Thanks for the advice. We have slightly modified that sentence in the abstract accordingly (Line 17): *“USPs constitute the largest family of de-ubiquitinases in cells having in common their proteolytic activity towards ubiquitin-modified substrates, except for USPL1, which is a distant member the USP family with a unique preference for SUMO”*.

Line 35: an isopeptide bond

R. Sorry for that typo. We have corrected it.

Line 42: DUB families

R. Sorry for that typo. We have corrected it.

Line 44: in humans

R. Sorry for that. We have corrected it.

Line 65: Full-length USPL1...

R. Sorry for that. We have corrected it.

Line 67 where it is...

R. Sorry for that. We have corrected it.

Line 80: avoid term “ubiquitin USPs” for something more precise: USPs that deubiquitinate

R. Thanks for the advice. We have modified that sentence accordingly (Line 80): *“The sequence of the catalytic domain of USPL1 is unequivocally related to the USPs with deubiquitinase activity, which probably forms the “canonical” right hand-like structure with palm, thumb and fingers subdomains”.*

Line 94: USP ubiquitin specific protease family:

R. Thanks for the advice. We have added the term “specific” to that sentence (Line 94): *“The catalytic domain for recombinant expression of human USPL1 is based on the sequence alignment with members of the USP ubiquitin specific protease family, and comprises from residue Met212 to Leu514 (Sup. Figure 1)”.*

Line 98: clarify: are USP7 and USP28 the closest related USP members with the highest shared sequence identity to USPL1 based on the catalytic domain? Clarify which USP has the highest shared sequence identity with USPL1.

R. The sequence identity between USPL1 with all other members of the USP family varies from 13% to 17%, being USP7 one of the highest. These numbers can vary 1 or 2 points depending on the starting and ending point of the USP catalytic domain, and should be taken in consideration other factors such as homology and presence and length of gaps in the alignment. We have modified that sentence in the text to clarify this point (Line 98): *“The USPL1 sequence identity is one of the lowest in the USP family, with identities ranging from 16% for USP7 to 14% for USP28 as examples.”*

Line 134: specify the crystallisation conditions in the text

R. Following the reviewer's suggestion, we have added the crystallization condition in the main text (Line 134): *“... were obtained in the initial screening in a condition containing 0.2 M potassium thiocyanate, 0.1 M sodium acetate 5.5, 8% w/v PEG20000 and 8% w/v PEG500MME”.*

Line 137: please replace “crystal phases” with a more accurate term.

R. Thanks for the advice. We have replaced the term “crystal phases” by the term “structure” (Line 137): *“... but fortunately we were able to solve the structure by using the recently reported USPL1 model from the alpha-fold server”.*

Line 162 and elsewhere: replace “ubiquitin USP members” with “USP members acting on ubiquitinated substrates” or similar as the term ubiquitin USP is confusing (it equals: ubiquitin ubiquitin specific protease).

R. Thanks for the advice. We have replaced “ubiquitin USP members” for “USPs with deubiquitinase activity” in the text (Line 162). *“As mentioned before, the sequence identity between USPL1 and the other ~~ubiquitin~~ USP members with deubiquitinase activity is only around 15%,...”.*

Line 173: "...the sequence identity is difficult to predict"; it shouldn't be as the stretches of sequence in these regions can still be aligned even if structures do not align well? Please be precise

R. Sorry for the confusing sentence. We have removed that sentence (Line 173).

Line 181 and elsewhere in the manuscript: the lack of the blocking loop.... Two active site loops are generally referred to as blocking Loop 1 and blocking Loop 2 in USPs, respectively. Please specify which of the two blocking loops is missing (or shorter?) in USPL1.

R. We have now better specified in the text the lack of the two "Blocking Loops (1 & 2)" or a significant part of those loops are missing in the sequence and structure of USPL1, when compared with other USP family members, as shown in Figure 3A. We have added a new sentence in its reference legend: "... *Three catalytic residues are depicted in red. Active site "Blocking loops (1 & 2)" are denoted by red squares. All images were prepared with PYMOL.*" Also in the main text we have specified the absence of the two blocking loops in the active site (Line 181 and elsewhere).

Line 199 and elsewhere: add reference and PDB code for where direct structural comparisons with published structures are described. Here: USP28-Ubiquitin complex structure.

R. Sorry for that. We have added the PDB code and reference to the USP28-ubiquitin (Line 199) and elsewhere in the text.

Line 253: "a subtle rigid body adjustment to fix ubiquitin": please rephrase: in some USPs I wouldn't describe the conformational changes between free state and distal ubiquitin bound state in this area as subtle, these can reach several Å: for example, in USP12 the fingers domain opens up by over 8 Å. Also, for USP8 significant movement of the fingers region is anticipated to fit a ubiquitinated substrate. please revise.

R. We have rephrased the sentence and removed the term "subtle" (Line 253).

Line 254: "the USPL1 the orientation of the fingers subdomain seems already fixed in the apo form": on which basis has this statement been made? There is no USPL1 structure in the free form available to justify this. Please revise. "As a consequence, the fingers subdomain interface with SUMO2 is unique in USPL1 and do not share any contact with ubiquitin USPs complexes": please clarify the meaning of this. I suspect it is mainly about the observed orientation of the fingers domain with regards to the core of SUMO but this needs more detail for clarification.

R. This statement about the apo form of USPL1 is based on the AlphaFold-2 model that we have used to solve the phases of our diffraction data by molecular replacement, which basically proves the quality of the model prediction. Right now, I think that there is enough evidence of acceptance of AlphaFold-2 models, however, since we don't have experimental evidence, we have rephrased the sentence (Line 254): "*However, in the USPL1-SUMO2 complex the orientation of the fingers subdomain is similar to the alphafold-2 model of the apo form (Figure 2) and displays a different orientation compared to all known structures of ubiquitin USPs,...*"

R. As suggested, we have revised the sentence (Line 258) "*the fingers subdomain interface with SUMO2 is unique in USPL1 and do not share any contact with ubiquitin USPs complexes*" for "*the contacts observed in the fingers subdomain interface with SUMO2 are*

not observed in the other USP-ubiquitin complexes, basically by the different orientation of the fingers subdomain with regards to SUMO2”.

Line 279, 280: the C-terminus... affinity for

R. Sorry for that. We have corrected both typos.

Line 301: in vitro in italics

R. Sorry for that. We have corrected that.

Line 377: rule out instead of discard

R. Thanks for the suggestion, we have replaced the word.

Table 1: needs more detail and be better annotated in places: what values exactly are quoted and what is shown in brackets. Show average B-factors for protease and SUMO-2 and waters separately. Include information on Ramachandran plot regions.

R. We have modified Table 1 accordingly, including the B-factors and the Ramachandran information of the final model.

Method section: pET instead of PET; which buffer was the protein in for crystallisations, please add.

R. We have corrected the plasmid pET nomenclature..

R. We have added the buffer condition for crystallization in the methods section (Line 686): “...in a buffer containing 20mM Tris 8.0, 170mM NaCl and 1mM DTT”.

Line 710: which SYPRO stain specifically?

R. We have added “SYPRO Ruby protein gel stain” in the methods (Line 710).

Discussion: the discussion is too narrow. What is missing are wider structural comparisons. Please include a paragraph on whether there are any shared characteristics with the SENP family of proteases. It is likely that there are some similarities despite the structural differences.

R. Thanks for the suggestion. The structural comparison of the inhibition mechanism with members of the SENP/ULP family is poor. Specific contacts with SUMO2 are not shared between the two protease families. Only, the binding of the C-terminus and a region in the *thumb* subdomain of USPL1 is shared with SENP2, but the different specific contacts.

As suggested, we have included a new paragraph regarding the USPL1 structural comparison with the SENP/ULP family at the end of the discussion: “*It is also interesting to compare the SUMO interface between USPL1 and members of the SUMO-specific SENP/ULP protease family. Interestingly, in this family...*”

Figure 6: Error bars have to be shown for the time courses and the number or replicates or repeats stated in order to evaluate the statistical significance.

R. We have prepared a table in figure 6d for to show the statistical significance of the ubiquitin-AMC fluorescence time course reactions of the point mutants. In this table we show the mean value of the initial slope of the saturation time course curves, as well as its corresponding standard deviation and the statistical significance (t-test).

Suppl. Fig. 1b: On what basis have USP sequences shown be selected? This needs to be clearly stated.

R. We have selected 5 different USP sequences to compare with USPL1 in the Supplementary Fig.1b. USP7 was selected as the “canonical” USP catalytic domain, since it was the first member of the family characterized in detail. USP8 was selected because in our initial prediction at the beginning of the project displayed the highest structural similarity. USP18 was selected because it was specific for ISG15, not ubiquitin. USP25 and USP28 were selected because we worked previously with them. In any case, we did not want to populate the list with many sequences, so we chose only up to 5 different USP sequences. All of them showed between a 14-17% sequence identity with the catalytic domain of USPL1.

Reviewer #2 (Remarks to the Author):

By sequence, USPL1 is member of the so-called USP class of Ubiquitin-specific proteases. Nonetheless, however, it does not act upon ubiquitin but instead is specific for SUMO2. The authors report a crystal structure of covalently trapped complex between the catalytic domain (residues 212-514) of human USPL1 and the human precursor of SUMO2 at 1.8 Å resolution.

Thereby, important differences in the interaction of USPL1 with SUMO2 versus ubiquitin-specific members of the USP family and ubiquitin are revealed.

It also shown that USPL1 has a strong preference towards SUMO2 over SUMO1 or ubiquitin.

The structural data identified a number of interactions that could explain the unusual preference of USPL1, the relevance for the action upon SUMO2 is supported by a mutational analysis for several of them including Arg 324, Phe335, Asn398, Tyr451, Phe457, and His493 in biochemical assays.

Altogether, this appealing structure/function analysis illuminates an interesting evolutionary diversification of specificities towards distinct UbL family proteins starting from a conserved enzyme fold, that will be of interest to researchers in the field.

Specific comments on details:

There is a bit of a muddle with figure mentioning in the text:

Line 150: “Figure 1b,c” should probably be Figure 2b,c

R. Sorry for that. The figure number has been corrected for Figure2b,c.

Line 152: “Figure 1d” should probably be Figure 2d (but see below).

R. Sorry for that. The figure number has been corrected for Figure2c.

Figure 2D appears in the text (line 155) but neither in the figure nor its legend.

R. Sorry for that. The figure number has been corrected for Figure2c.

Line 164: Figure 2a, should probably be Figure 3a

R. Sorry for that. The figure number has been corrected for Figure3a.

It is concluded that M330 “does not seem critical for binding and catalysis.” However, even though the M330A mutant appears to bind better in the assay presented on figure 6b, it shows diminished processing of RanGAP1-SUMO2 (Figure 6e) and also reduced reaction with SUMO2-AMC as compared to wt. This may not be “critical” but at least seems significant.

R. Thanks for this observation. We have removed the term “critical” for “significant” in the main text. Line 282: “*and at least in USPL1, it only seems a bit significant for binding and catalysis*”.

Line 288: One should not call it “conserved histidine mutants”

R. Thanks for the advice. We have removed the term “mutants” for “residues”. Line 288: “*Finally, two conserved histidine residues were analyzed...*”.

Draft Only

REVIEWERS' COMMENTS

Reviewer #1 (Remarks to the Author):

The authors have adequately clarified and addressed the points previously raised and I am now happy to recommend publication of this work.

Draft Only